# River Green Land and Its Influence on Urban Economy, Leisure Development, Ecological Protection, and the Well-Being of the Elderly

Xiao-Juan Du [1], Hsiao-Hsien Lin [2], I-Cheng Hsu [3], Ying Ling [4,*], Su-Fang Zhang [5,*] and Qi-Yuan Li [6]

1 College of Sports Economics and Management, Guangxi University of Finance and Economics, Nanning 530003, China
2 Department of Leisure Industry Management, National Chin-Yi University of Technology, Taichung City 411030, Taiwan
3 Marketing and Distribution Management Department, Tzu Chi University of Science and Technology, Hualien 97005, Taiwan
4 School of General Education, Guangxi Vocational & Technical College, Nanning 530226, China
5 Department of Tourism Management, Athena Institute of Holistic Wellness, WuYi University, No 26, WuYi Avenue, Wuyishan 354300, China
6 School of Physical Education, Jiaying University, Meizhou 514015, China
* Correspondence: shen0909169168@gmail.com (Y.L.); tina060513015627929@gmail.com (S.-F.Z.)

**Abstract:** The purpose of this study is to analyze whether the green space generated by river water engineering can promote urban development and the well-being of the elderly in high-risk environments. Firstly, quantitative research methods were used to analyze 750 valid questionnaires, and IBM SPSS Statistics 26.0 statistical software was used for data inspection. We continued to adopt the qualitative research method and collect the opinions of nine respondents according to the results of the questionnaire analysis. The data were then integrated, categorized, summarized, compared, and finally examined by multivariate analysis. The river green land has rich ecological and natural landscape resources and spacious leisure space, creating a comfortable leisure and living environment. This can increase job opportunities, promote community and economic development, and create safe leisure conditions. It can also improve people's willingness for leisure, increase opportunities for interpersonal communication, and improve the quality of life and happiness of the elderly. If we should strengthen the soil structure of the river, pay attention to ecological conservation, and reduce exhaust gas and noise pollution to provide a more complete leisure space and enhance the contribution of river green space to rural areas and lives of the local elderly.

**Keywords:** environmental risks; water conservancy projects; river greening; urban sustainability; elderly; well-being

## 1. Introduction

With the advancement of time and strong economic development, population aging is becoming more and more serious worldwide [1]. Population aging refers the increasing proportion of elderly people aged 60 or 65 years and above in the total population [2]. Studies have shown that 11% of the world's population is currently 60 years or older, which is expected to increase by 22% by 2050 [3]. However, with the increase in the number of people over 60 years old, the impact of the accelerated aging phenomenon on economic and social development will increase as the overall population age structure changes [4]. Moreover, due to the health status and functional limitations of the elderly population, their sensory abilities are low, and their sensitivity to environmental stress is high, making them vulnerable to environmental impacts [5]. Therefore, in the process of human future

development and construction, finding ways to promote economic and industrial development, constructing a stable and healthy leisure environment, and enhancing the well-being of the elderly will become important issues.

Rivers are rich in water resources and have a complete water and ecological cycle system [6]. Since rivers have an inexhaustible supply of fresh water throughout the year, it can be used for drinking and irrigation. During the rainy season or when the river is full of water, it can provide people with water transportation. River water can meet people's needs for living, working, leisure, and commercial transportation, leading to economic benefits and improved quality of life [6,7]. Ultimately, it can provide humans with a safe and healthy leisure environment and a comfortable and happy well-being [8]. However, due to climate change, the amount of heavy rainfall has increased dramatically, and the capacity of rivers is insufficient, resulting in the collapse of river banks and causing flooding, which causes impacts to the surrounding ecological environment and human beings, endangering the safety of life and property [9] and destroying people's stable living environment [10]. We can see that although river water resources are beneficial to humans and rural development, there is no way to avoid the fact that river water causes flooding due to climate change, which persecutes human and rural development [9,10].

Water conservation is an engineering technique to improve the environment of rivers, reservoirs, or coasts. It is a public decision to buffer flood volume, strengthen river banks and surrounding soil, and prevent river collapse by excavating, dredging, and filling river banks or extending the space on both sides of the river inward [11]. It originally aimed to improve river siltation, strengthen river banks, and reduce flooding [9]. In addition to creating flood control, navigation, irrigation, and power generation, it reduces flooding, addresses water pollution, and provides clean water [12]. However, as people become more aware of recreational and tourism activities, and as environmental risks increase due to epidemics, the public loses space to maintain health and is threatened [5]. Therefore, there is an urgent need to find safe and open environments to allow the public to engage in leisure activities and improve their physical and mental health [13].

Green space refers to the space that is artificially planted with a large amount of vegetation. A river green land is a green environment in which the government uses the inward extension of a river to plant a large amount of vegetation or shrubs [13]. It can satisfy people to obtain flood control and dredging effect and maintain the natural environment and ecology. It can also provide people with leisure, recreation, entertainment, sports, and tourism activities [13] and has promoted people's willingness to participate in sports, maintain health, and create a happy city [14]. In summary, we believe that at this stage, river greenery can be developed towards the goals of flood control, ecological conservation, and human health promotion to achieve sustainable water resources.

The Mulan River is located in Fujian Province and flows through the Licheng and Chengxi districts, with a total length of 105 km. It is one of the important freshwater rivers in the area, bringing abundant water to villages and residents for livelihood and farming [15]. Due to climate change and abnormal rainfall, coupled with the inadequate technology of early water conservation projects [15], it is easy for the Mulan River to burst. The large amount of river water washes away the riverbank topography and destroys the soil structure, reducing the rural hinterland around the river, which impacts the local economy and ecological environment [16]. However, due to the improvement of water engineering technology and the local government's investment in improving sewage treatment plants and constructing ecological flood storage ponds [15,16], not only is the chance of flooding is effectively reduced, but the extensive green space is also increased and local development is promoted [17]. It has become a successful case of leisure, livability, vacation, natural ecology, history and culture, special tourism, and green energy industry, and it can enhance the attractiveness of living [18]. According to statistics, green areas can bring more than CNY 20 million of agricultural income to the local villages yearly, increase the GDP exceeding CNY 200 billion [19], and increase the population to 1,221,300 people [20]. It is evident that although the original natural environment of

rivers can nurture diverse ecology and provide abundant freshwater, the existing river channels and surrounding soil structure cannot cope with the abnormal rainfall [9], and the development of rural land can destroy the original river channels and ecology [10]. It needs to be improved by effective water conservation engineering decisions. Moreover, the derived river greenery can realize water treatment and maintain biodiversity. It can also provide stable water resources for irrigation and increase agricultural production and income [19]. It can also help enterprises to transform and develop new industries [18] and provide space for people to relax and travel, increasing leisure and consumption desire [8] and ultimately promoting rural development [19]. Therefore, we believe that the emergence of green areas enhances flood control, improves village construction, promotes development, and provides a comfortable and stable living environment for the elderly, increasing their sense of well-being.

However, some scholars have argued that the expected effects of decisions or theories are different from the actual benefits [21]. Different environmental, human, or non-human factors can be interfering factors to influence the effectiveness of development [18]. The results need to be verified over time, and the answers have to be obtained through the feelings of the actual participants [22]. Therefore, we believe the answer can be obtained by obtaining the perceptions of the elderly who live in the area or engage in leisure activities. In addition, the current literature on the river and water conservancy projects only discusses the current development of the village economy, village society, village environment, and natural ecological environment [18] and rarely discusses the impact on well-being at the same time [23]. The relationship among village development, nature, ecology, and well-being has not been analyzed simultaneously. There are no discussions on the effects of the river and water conservancy projects or green areas on the economic and ecological development of villages and the well-being of the elderly.

The purpose of this study is to analyze the effects of river green spaces on rural economic, social, and environmental development, nature conservation, and the well-being of the elderly through the leisure experiences and perceptions of local elderly people in the Mulan River area. We believe that the main purpose of this investigation is to understand the importance of river green spaces for rural development and river ecology conservation and the role of the elderly in regulating their perceptions of well-being in green space activities. The results can be used to appeal to people and other local governments to strengthen the importance of utilizing river resources and revitalizing green spaces. This is to promote the sustainable development of water resources, the elderly, and ecological environment, enhance well-being, and achieve the goal of a water recycling economy.

## 2. Literature Review

### 2.1. Well-Being of the Elderly

Well-being is a complex and subjective psychological attitude [24]. It is based on an individual's subjective feelings, combined with the feelings of one's own preferences, to measure the degree of an individual's current well-being indicators [25]. The well-being of the elderly is achieved when something satisfies their physiological needs, and they can obtain joy and happiness from the interactive atmosphere of the external environment [26].

Early studies found that well-being was measured by economic growth [27], but later studies concluded that a single economic dimension could not accurately measure well-being and needs to include personal, psychological, and external states [28] to be considered a complete indicator of well-being. Scholars have suggested that the current well-being of an individual can be determined by constructing a retirement environment, relaxing physically and mentally, living a more optimistic life, using different skills and abilities, enjoying life, and interacting socially [29]. Therefore, we believe that the assessment of constructing a retirement environment, relaxing physically and mentally, living a more optimistic life, using different skills and abilities, liking their life, and social interactions can predict the well-being of the elderly.

Some scholars pointed out that river water conservancy projects have the benefits of leisure, entertainment, tourism, and livability after construction [30]. Therefore, they believed that elderly people should engage in activities around the river greenery to obtain spiritual comfort and increase the pleasure of life [31], stabilize their emotions, and obtain a high quality of life [32]. Thus, the construction of water conservancy projects should positively affect the well-being of the elderly. Therefore, we believe that river water conservancy projects positively affect the well-being of the elderly.

### 2.2. Village Tourism Development

Tourism is a mode of consumption behavior that uses local human, natural, and characteristic industries as tourism resources to attract people to participate in activities in a specific area [33]. Tourism development is the promotion of local tourism industries or activities with the help of tourism resources to promote local development [34]. Village tourism development is a phenomenon in which villages use tourism resources to promote tourism industries or activities to revitalize the local economy, promote social harmony, and improve the community environment [35].

Scholars believe that water conservancy projects can reduce the chance of flooding, stabilize freshwater resources, and protect the ecological diversity of rivers [36]. However, the emergence of green space should have more functions than just enhancing the dredging effect and protecting the ecology. For example, green space can beautify the community environment and increase development space and entrepreneurial opportunities [8]. This can promote rural economic development, improve the community living environment, and indirectly enhance community health and environmental quality [37]. It can bring a safe living and leisure environment for the elderly and improve their sense of well-being [29]. It can be seen that water conservancy projects contribute to river dredging and flood reduction, but also the resulting green space should also provide sufficient space for development. This can improve the living environment, promote economic development, and ultimately create a diverse, comfortable, and safe living space to enhance the well-being of the elderly.

However, expected travel decisions and actual outcomes are susceptible to positive and negative effects [38]. In addition, older adults have poorer physical and psychological qualities, high environmental sensitivity, and severe anxiety problems [39]. Differences in the quality and effectiveness of the travel environment have varying degrees of impact on older adults' lives and travel experiences. Scholars have suggested that the impact of river engineering green spaces on rural development can be understood by exploring the current state of economic, social, and environmental development in rural areas [40]. Therefore, we believe that the impact of river green spaces on village development can be understood through public perceptions of the current state of economic, social, and environmental development in rural areas.

### 2.3. Natural Environment and Ecology

The natural environment is an environment that is only indirectly or slightly affected by human beings, and the original natural appearance has not changed significantly [35]. The ecological environment is composed of ecological relationships, the sum of various ecological factors and ecological relationships on which living organisms depend for survival, development, reproduction, and evolution [36]. Natural environment and ecological development refer to the changes in the original natural environment and ecology due to external or internal disturbance factors [32].

Some scholars believe that rivers have abundant water resources, rich environmental resources, and ecological diversity [30], and with the planning of water conservancy projects, they should provide a safe and comfortable living and leisure environment [37]. In addition to improving the flooding problem of rivers through water conservancy projects, the project can help promote the positive development of the natural environment and ecology [36]. Moreover, conservation policies can raise people's awareness of conservation

and slow down the rate of natural environment destruction [31]. The natural environment and ecology can be used to provide a comfortable living environment for the elderly, providing them with emotional relief and improving their quality of life [21]. However, promoting tourism development may also cause river pollution and interfere with water and ecology [18]. Therefore, scholars have pointed out that if we can understand the current state of the natural environment and ecology, we can know the impact of water conservancy projects on the current state of the natural environment and ecological development around the villages [36]. Therefore, we believe that the influence of the river water conservation project can be understood if the natural environment and ecological development are examined in their current state.

*2.4. Impact of River and Water Conservancy Projects on Village Development, Natural Environment, and Ecology on the Well-Being of the Elderly*

According to scholars, river conservancy projects were originally used to deal with the collection, storage, control, transportation, regulation, use, and protection of the surface of water resources [23] and are one of the ways for stable human use of river water resources. River conservancy projects should also create conditions to enhance local economic conditions, increase employment opportunities, enhance economic returns, improve the quality of life [21], and enhance the well-being of the elderly [30].

While a well-developed economy, diversified leisure resources, and a safe living atmosphere can create a livable environment [21], a well-developed transportation system, and medical facilities which can improve the community's public health environment and quality of life [21]. Diversity of ecology and a good natural environment [38] can provide comfortable living spaces. All of these are beneficial for the elderly to relax and improve their sense of well-being [24]. Thus, it can be concluded that water conservation projects should be beneficial to promote the development of villages and natural ecology which can indirectly promote the well-being of the elderly.

Therefore, based on the above inferences, we believe that if water conservancy projects can improve the river, it should be beneficial to improve the current situation of the village and ecological development and increase the well-being of the elderly.

## 3. Methodology

*3.1. Framework and Hypothesis*

This study was conducted by using a mixed research method to investigate the issues of village development [33–40], natural environment and ecological development [41–43], and well-being [24–30] of the elderly in the villages surrounding Mulan Creek. We analyzed whether river water projects could improve the village economy, construct a friendly leisure environment, and promote the physical and mental health of the elderly. The structure of this study is shown in Figure 1.

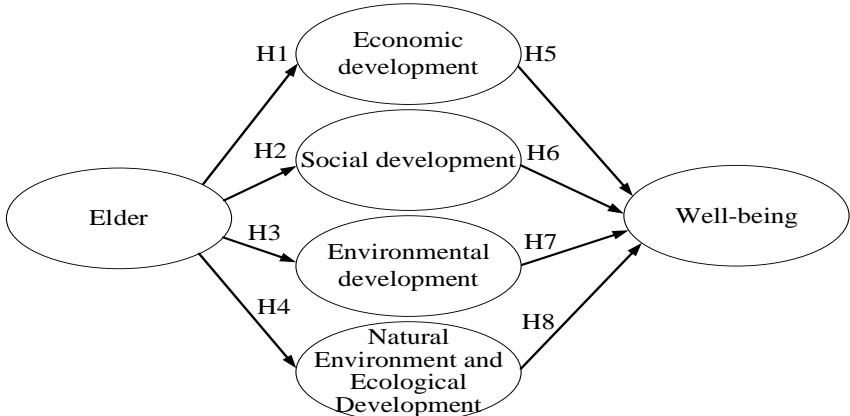

**Figure 1.** Research framework.

*3.2. Hypothesis*

Some scholars believe that rivers have abundant water resources [18], and the government uses rivers to build water conservancy projects to promote the development of surrounding villages [17,18]. The key to village development is to make good development decisions and to obtain people's support [21]. The elderly are often a large group of people who stay in villages [22], so getting sufficient resources to improve the living environment of the elderly is beneficial to promote village development and create a comfortable and pleasant living atmosphere [37], which in turn improves the confidence of the elderly in government decisions [22]. However, due to the cognitive differences of the elderly, different views will be derived, which will also affect the willingness to participate in decision making [8], village development, and natural ecological development and reduce the well-being of the elderly [24]. Therefore, we believe the river water conservancy project should cause differences in the elderly's perceptions of village development (economic, social, and environmental), natural environment, and ecological development in the surrounding villages. In addition, village development (economic, social, and environmental), natural environment, and eco-logical development perceptions may also have an impact on the well-being of the elderly. We proposed the following 8 hypotheses for testing based on the above inferences.

**Hypothesis 1 (H1):** *The elderly people think that green areas do not help to improve the economic development of rural areas.*

**Hypothesis 2 (H2):** *The elderly think that green areas do not help improve the current situation of rural social development.*

**Hypothesis 3 (H3):** *The elderly people think that green areas do not help to improve the economic development of the villages.*

**Hypothesis 4 (H4):** *The elderly think green areas do not help improve the natural environment and ecological development around the villages.*

**Hypothesis 5 (H5):** *The current economic development of the villages helps to improve the happiness of the elderly.*

**Hypothesis 6 (H6):** *The current state of social development in rural areas helps to improve the sense of well-being of the elderly.*

**Hypothesis 7 (H7):** *The current state of rural environmental development helps to enhance the happiness of the elderly.*

**Hypothesis 8 (H8):** *The current state of the natural environment and ecological development around the villages helps to enhance the happiness of the elderly.*

*3.3. Research Process, Methods, Tools Design, and Analysis*

We first read the literature on the Mulan River and river conservancy projects, village development, natural environment, ecological development, and well-being to verify the research themes. Then, we edited the questionnaire by referring to the literature on river water conservancy projects [11–14], village development [33–40], natural environment and ecological development [41], and well-being [24–32]. After completing the preliminary questionnaire, some scholars were invited to conduct a content validity check to validate the questionnaire's content. A pre-test of 100 questionnaires was conducted in December 2021. SPSS 26.0 statistical software was used to check the validity of the questionnaire, and the official questionnaire was edited after adopting the questionnaire questions with alpha values greater than 0.6.

A further 800 official questionnaires were distributed in January–February 2022, and 750 valid questionnaires (93.7%) were finally obtained. Then, based on the results of the data analysis, we interviewed experts, villagers, tourists, and business owners. Finally, all the data were compiled, categorized, summarized, and compared for data processing, and multivariate check analysis was used. The analysis of the questionnaire and the description of the respondents' background information are shown in Table 1 below.

**Table 1.** Questionnaire and Respondent Background Information.

| Facet ($\alpha$) | | Issue ($\alpha$) | | KMO | Bartlett ($\chi^2$) | df | *p* |
|---|---|---|---|---|---|---|---|
| Village development (0.942) | Economy (0.939) | E1: Increase tourism construction and industry (0.929)<br>E2: Combination of local characteristics and industries (0.934)<br>E3: Maintain village public facilities (0.927)<br>E4: Increased entrepreneurship and employment opportunities (0.927)<br>E5: Increase salary income (0.928)<br>E6: Improve village transportation planning (0.926)<br>E7: Increased popularity (0.929)<br>E8: Develop village protection policies (0.930) | | 0.913 | 2369.707 | 28 | 0.000 |
| | Society (0.919) | S1: Improve village's characteristic culture (0.900)<br>S2: Improving the quality of village tourism services (0.911)<br>S3: Increased leisure opportunities in villages (0.905)<br>S4: Public participation in village development (0.905)<br>S5: Increase the friendly and interactive atmosphere of people (0.903)<br>S6: Increase the quality of police and security personnel (0.911)<br>S7: Improve the willingness of youth to return to the village for development (0.905) | | 0.903 | 1702.696 | 21 | 0.000 |
| | Environment (0.891) | Em1: The living environment is affected by tourists (0.884)<br>Em2: It is more convenient to take transportation (0.874)<br>Em3: Increased exhaust gas and noise pollution from motor vehicles (0.874)<br>Em4: The public trash can is clearly set and sufficient (0.860)<br>Em5: Public toilets are obvious and sufficient (0.864)<br>Em6: Well-maintained village landscape facilities and historic sites (0.870) 0.873 | | 0.873 | 1199.103 | 15 | 0.000 |
| Natural Environment and Ecology (0.886) | | N1: Turbidity of river water quality (0.863)<br>N2: Change in soil structure of the river (0.841)<br>N3: Decrease in the number of ecological species (0.868)<br>N4: Damage to river ecology and natural environment (0.875)<br>N5: Disposal of tourist garbage (0.855) | | 0.856 | 1002.474 | 10 | 0.000 |
| Well-being (0.897) | | W1: Sense of participation in life (0.877)<br>W2: Construct retirement living environment (0.879)<br>W3: Physical and mental relaxation (0.875)<br>W4: Life is more optimistic (0.882)<br>W5: Use of different skills and abilities (0.887)<br>W6: Like my life (0.879)<br>W7: Social interaction (0.881) | | 0.881 | 1399.012 | 21 | 0.000 |

| | code | gender | specialty | age | code | gender | specialty | age |
|---|---|---|---|---|---|---|---|---|
| Professionals or interviewees | P1 | male | tourist decision-making | 42 | G3 | male | tour guide | 42 |
| | P2 | female | travel management | 45 | G4 | male | tour guide | 45 |
| | P3 | male | event management | 49 | O1 | female | hotelier operator | 55 |
| | P4 | female | public health | 47 | O2 | female | restaurant operator | 62 |
| | G1 | female | tour guide | 40 | E1 | male | elder | 66 |
| | G2 | female | tour guide | 38 | E2 | female | elder | 65 |

### 3.4. Scope, Object, and Limitations

The Mulan River is one of the important rivers in Fujian Province, China [42]. Although flooding once devastated the village, it has been improved by the government's active river improvement. The river is rich in water resources and ecological diversity and provides water for human life and agricultural, fishery, and industrial development [23]. In addition, the government has actively invested in improving existing public facilities, improving

the quality of water resources, raising public awareness of environmental protection, and utilizing local tourism resources to promote village development [30], which has contributed to an effective increase in the number of elderly people in the area, reaching 180,200 [20]. The improvement of the Mulan River has created rich leisure and tourism resources and a safe living environment for the local community [8], which is conducive to improving the quality of life of the elderly [30,40]. Therefore, we believe that using the elderly in the villages around the Mulan River as the survey target is representative.

However, even with the large number of elderly people in this study and the use of convenience sampling, online questionnaire platform, and on-site sampling, the impact of viruses on the health of the elderly is still overwhelming due to the decline of their physiological and psychological functions [5]. In addition, the local epidemic status in China is still unstable, and the research team has insufficient human resources, resources, and funding. The data and survey results may be influenced by these factors. Therefore, we believe that these factors may affect the study results. Therefore, the limitations and difficulties mentioned above are listed as suggestions for follow-up studies, and we will rely on subsequent researchers to improve them.

### 3.5. Ethical Considerations

The manuscript first used a literature review to identify the proposed research themes and instruments. Then, a content validity check was applied to analyze the reliability to confirm the validity of the questionnaire content. Secondly, during the sampling process, the research assistant identified himself/herself and explained his/her purpose, then started the survey and confirmed the authenticity of the interviewed content after the respondents agreed. Finally, all data were summarized, categorized, compared, and analyzed from a multi-viewpoint perspective [44,45]. Therefore, this research design and manuscript content complied with the restrictions of Administrative Circular No. 1010265075 of the Department of Health, Executive Yuan, Taiwan [46], and followed the regulations of Article 1004 and Article 1009 of the Civil Code of China [47]. It was designed in compliance with the regulations and the principles of fairness, openness, and equity [48,49]. Therefore, we believe that the research process is consistent with the principles of the Declaration of Helsinki.

## 4. Results

This study was conducted with 750 valid questionnaires. The analysis revealed that 380 (50.7%) of the sample population were male and 370 (49.3%) were female. The sample population's background and the current status of the villages' economic, social, environmental, natural environment, and ecological development and their effects on well-being were then investigated using basic statistical tests and Pearson's performance correlation analysis.

### 4.1. Cognition of the Status Quo of the Village Economy, Society, Environment, and Natural Ecological Development

The analysis found that in terms of the village economy, seniors felt the most strongly about increasing their income (3.97) and the least strongly about establishing special industry integration (3.62). In terms of village society, seniors felt the most strongly about increasing people's friendly and interactive atmosphere (3.97) and the least about improving the quality of village tourism services (3.62). In terms of the village environment, the elderly felt the most strongly about the convenience of transportation, the obvious and sufficient provision of public garbage cans, and the obvious and sufficient provision of public toilets (3.94), and the least about the increase of exhaust and noise pollution from motor vehicles (3.69). Regarding the natural environment and ecological development, the elderly felt the highest about soil erosion along the river, change of original ecological habitat, and increase in tourism waste (3.94) and the lowest about river turbidity (3.69), as shown in Table 2.

**Table 2.** Cognitive analysis of the current situation of village economic, social, environmental, and natural ecological development.

| Facet (α) | Issue (α) | M | Ranking | Facet (α) | Issue (α) | M | Ranking |
|---|---|---|---|---|---|---|---|
| Economy | Increase tourism construction and industry | 3.78 | 5 | Society | Improve village characteristic culture | 3.69 | 6 |
| | Combination of local characteristics and industries | 3.64 | 7 | | Improving the quality of village tourism services | 3.62 | 7 |
| | Maintain village public facilities | 3.67 | 6 | | Increased leisure opportunities in villages | 3.86 | 4 |
| | Increased entrepreneurship and employment opportunities | 3.84 | 2 | | Public participation in village development | 3.82 | 5 |
| | Increase salary income | 3.88 | 1 | | Increase the friendly and interactive atmosphere of people | 3.97 | 1 |
| | Improve village transportation planning | 3.84 | 3 | | Increase the quality of police and security personnel | 3.91 | 3 |
| | Increased popularity | 3.85 | 2 | | Improve the willingness of youth to return to the village for development | 3.96 | 2 |
| | Develop village protection policies | 3.81 | 4 | | | | |
| Environment | The living environment is affected by tourists | 3.81 | 2 | Natural Environment and Ecology | Turbidity of river water quality | 3.69 | 3 |
| | It is more convenient to take transportation | 3.94 | 1 | | Changes in the soil structure of the river | 3.94 | 1 |
| | Increased exhaust gas and noise pollution from motor vehicles | 3.69 | 3 | | Decrease in the number of ecological species | 3.91 | 2 |
| | The public trash can is clearly set and sufficient | 3.94 | 1 | | Damage to river ecology and natural environment | 3.94 | 1 |
| | Public toilets are obvious and sufficient | 3.94 | 1 | | Littering of tourist garbage | 3.94 | 1 |
| | Well-maintained village landscape facilities and historic sites | 3.81 | 2 | | | | |

The results proved to be inconsistent with research Hypotheses 1–4. However, the water conservancy project still cannot create special industries and improve the quality of tourism services, and it still generates a large amount of tourism waste, causing soil erosion and destroying ecological habitats. The results proved inconsistent with research Hypotheses 1–4 and not completely consistent with the literature [38].

*4.2. Correlation Analysis of Village Economy, Society, Environment, Natural Environment, and Ecological Development on Happiness Cognition*

The correlations between the village's economic, social, environmental, natural environment, and ecological development and the well-being of the elderly were analyzed using Pearson's performance difference correlation analysis, as shown in Table 3. We found significant differences ($p < 0.01$) between the village's economic, social, environmental, natural environment, ecological development, and well-being. The analysis revealed that using different skills and abilities (0.518), constructing a retirement environment (0.332), constructing a retirement environment (0.301), and establishing a retirement living environment (0.252) had the greatest effect on well-being. Feeling involved in life (0.269), establishing a retirement living environment (0.252), and liking my life (0.135) were the least important, with liking my life (0.132) and feeling involved in life (0.119) having the least effect on well-being.

The results showed that the effectiveness of the economic, social, environmental, natural environment, and ecological development of villages influenced the well-being of the elderly. The results confirmed the establishment of Hypotheses 5 to 8, consistent with the literature [38,39]. Among them, different skills and abilities and constructing a retirement living environment had the highest influence, while liking my life and sense

of life participation had the lowest influence. Additionally, the better the village's economic, social, environmental, natural, and ecological development, the higher the sense of well-being.

**Table 3.** Analysis of the influence of village economy, society, environment, natural environment, and ecological development on well-being.

| Facet | Economy | Society | Environment | Natural Environment and Ecology |
|---|---|---|---|---|
| Well-being | 0.479 ** | 0.234 ** | 0.258 ** | 0.276 ** |
| Sense of participation in the life | 0.269 ** | 0.093 | 0.094 | 0.119 * |
| Construct a retirement living environment | 0.370 ** | 0.252 ** | 0.301 ** | 0.332 ** |
| Physical and mental relaxation | 0.324 ** | 0.171 ** | 0.198 ** | 0.247 ** |
| Life is more optimistic | 0.370 ** | 0.140 ** | 0.200 ** | 0.227 ** |
| Use of different skills and abilities | 0.518 ** | 0.239 ** | 0.249 ** | 0.235 ** |
| Like my life | 0.328 ** | 0.132 * | 0.135 ** | 0.152 ** |
| Social interaction | 0.444 ** | 0.242 ** | 0.226 ** | 0.197 ** |

Note: * $p < 0.05$, ** $p < 0.01$.

## 5. Discussion

### 5.1. Effects of River Water Conservancy Projects and Green Land on Village Economy, Society, Environment, Natural Environment, and Ecology

We believe that green space can be used by the government to develop recreational space and to develop tourism, as the implementation of water projects can help reduce flooding. However, the available tourism resources generated by the green space are limited, and the nature of the leisure activities or tourism developed is different from the local rural culture, industrial characteristics, and people's living habits. In addition, there are barriers to communication between relevant government departments. Therefore, even though green areas can promote rural development and improve economic efficiency and salary structure, because the government's development and resource integration is inefficient, there are few leisure or tourism activities. As a result, only the economic efficiency and salary structure can be improved, and it does not help to promote the goal of integrating green areas with local industries and tourism resources. This proved that the research results are inconsistent with the literature [32–37] and verified that research Hypothesis 1 is invalid.

In addition, when the government is willing to invest in water conservation projects to improve river problems, the resulting green space can be used for rural improvements or public facilities. This can create a harmonious living circle and enhance public communication and care opportunities for the disadvantaged. However, there is little space for green space development and insufficient entrepreneurial skills for residents, resulting in a single type of tourism activity and few leisure options for the public to invest in. In addition, the elderly have low spending power but high demand for quality services. As a result, although green areas can provide rural villages with the opportunity to improve the problem of inadequate public facilities and create an environment for public interaction, they still cannot satisfy the needs of the elderly to engage in leisure or to provide them with services. Nevertheless, it is still unable to meet the demand of the elderly for leisure or tourism activities. Therefore, this proves that the study results are inconsistent with the literature [33–39] and verifies that research Hypothesis 2 is invalid.

The green area has a large area of vegetation, a large space, a good view, good air, and a lot of space for development. This is beneficial for the government to add new public facilities and improve the current situation of joint traffic planning. It is also conducive to promoting tourism development in rural areas and solving the problem of insufficient public toilets and garbage bins. In addition, the Chinese government is actively promoting the carbon reduction policy and using electric vehicles to provide feeder services. Therefore, we believe it is beneficial to enhance the leisure desire of the elderly, stimulate consumption, and promote the benefits of tourism and rural economic development. This

has led to the belief that green areas can promote tourism and industrial development and that the development can improve local transportation and public facilities to meet the transportation needs of the elderly. Therefore, this proves that the study's results are inconsistent with the literature [33,42] and also verifies that research Hypothesis 3 is not valid.

Finally, green areas can preserve a large amount of vegetation, reduce the damage to the soil structure of rivers and maintain water quality, and help create a biodiverse environment. However, the large demand for rural construction and development and the large space for construction will easily interfere with the ecological conservation areas. In addition, tourism development attracts a large number of tourists, which will generate tourism waste and damage the environment. As a result, even though the green areas can maintain the benefits of flood control, the development still causes the elderly to have doubts about the quality of river water and the environment. Therefore, this proves that the results of the study are not consistent with the literature [33–38] and verifies that research Hypothesis 4 is not valid.

*5.2. The Impact of Village Economic, Social, Environmental, Natural Environment, and Ecological Development on Well-Being*

Regarding the effect of village economic development on the well-being of the elderly, we believe that the construction of river and water projects, the addition of tourism elements, the promotion of the surrounding villages, the injection of new industries, and increased investment further enhance the economic development of the villages; this enables the elderly to have a safe living environment and to improve their different living skills and abilities. However, as tourism continues to evolve along with technological development, the elderly are unable to fully handle the content of current tourism activities and familiarize themselves with the supporting software facilities due to the decline in their physical and mental functions. Therefore, seniors believe that economic development can enhance different skills and abilities, but it does not help increase their sense of participation. Therefore, this proves that Hypothesis 5 is not valid.

With regard to the impact of social and environmental development on the well-being of the elderly, village tourism and economic development have been developed after river training. The government has also improved the public facilities in the village by promoting water conservancy projects and the advantages of the surrounding green areas. In addition to improving medical care and access to medical facilities, the government has also introduced sufficient sanitation facilities to beautify the village living environment. Therefore, this proves that Hypothesis 6 is not valid.

However, it is not easy for the elderly to operate high-tech soft and hard facilities. Tourism development has led to increased garbage and tourism waste, and water and air pollution have followed. As a result, the elderly have a poor experience and are easily dissatisfied with their lives. Therefore, the elderly believe that although the village society and environment have improved, a safe and convenient living environment can be provided to facilitate a high quality of life after retirement. However, the convenience of travel was not improved, and pollution increased, which reduces life satisfaction and happiness. Therefore, this proves that Hypothesis 7 is not valid.

Regarding the effect of the natural environment and ecological development on the happiness of the elderly, the water conservancy project has transformed the village and provided spacious green space, promoted the tourism industry, and established local tourism characteristics. Therefore, the elderly believe that the water conservancy project is beneficial to improving the natural environment and ecological conditions in the community and around the river. However, due to the declining physiological functions of the elderly, they are less active and mobile, so they are less likely to go out for leisure and less willing to do so. As a result, they are unable to appreciate the benefits and impact that can be achieved in their daily lives when the natural environment and ecology are improved. Therefore, the elderly believe that even if the surrounding natural environment and ecology are improved,

it will only increase the benefits of high quality of life after retirement, but it will not help to increase the sense of participation in life. Therefore, this proves that Hypothesis 8 is not valid.

## 6. Conclusions

We believe that river green land has the ability to create a comfortable and safe leisure environment due to its rich ecological resources, diverse natural landscapes, and spacious leisure space. This increases job opportunities, promotes community and economic development, and creates safe living spaces. This has the potential to increase opportunities for mass interaction and improve the quality of life and happiness of the elderly. However, management issues have not yet been resolved due to over-development, damage to the soil structure of the river, changes in the natural environment and ecology, and serious waste and gas pollution. As a result, the elderly believe that although green land promotes rural development and improves the quality of existing materials and the environment, it still cannot truly meet their needs and enhance their sense of well-being.

We propose the following recommendations based on the study limitations and analysis results. The wisdom and experience of the elderly can be utilized to cultivate in-depth guided tours and interpretation services, creating work opportunities and leisure features. The village's existing humanities, arts, and architectural features can be used to develop different experiential activities and regular training should be held to enhance the quality and reputation of tourism services by adding professional knowledge. In addition, governments should promote residents' willingness to participate in village development decisions in order to enhance policy cooperation and recognition. Local culture and characteristics should be used to organize cultural exchange activities in villages to promote interaction among people. There should be a reduction in water and air pollution, improvement in the living environment, and a joint building of a friendly and safe environment for leisure activities and tourism for the elderly to increase the desire to live and travel. Finally, the research topic was only focused on the elderly in the villages around Mulan River, which has limited human, material and financial resources. We suggest that subsequent researchers extend the discussion to other water areas, such as lakes and reservoirs. The impact of civil engineering works in mountainous woodland and swamp areas should be analyzed. Different research methods, analysis methods, or increased sample sizes should be used for the analysis. The discussion should be extend to the feelings of different backgrounds and the impact of different issues.

**Author Contributions:** Conceptualization, X.-J.D. and H.-H.L.; Methodology, X.-J.D. and H.-H.L.; Software, I.-C.H.; Validation, I.-C.H. and Y.L.; Formal analysis, H.-H.L. and Y.L.; Investigation, I.-C.H.; Resources, I.-C.H.; Data curation, Q.-Y.L.; Writing—original draft, Y.L.; Writing—review & editing, S.-F.Z.; Visualization, Y.L.; Supervision, X.-J.D.; Project administration, X.-J.D. and S.-F.Z.; Funding acquisition, Q.-Y.L. All authors have read and agreed to the published version of the manuscript.

**Funding:** This research received no external funding.

**Institutional Review Board Statement:** This research design and manuscript content complied with the restrictions of Administrative Circular No. 1010265075 of the Department of Health, Executive Yuan, Taiwan [46], and followed the regulations of Article 1004 and Article 1009 of the Civil Code of China [47]. It was designed in compliance with the regulations and the principles of fairness, openness, and equity [48,49]. Therefore, we believe that the research process is consistent with the principles of the Declaration of Helsinki.

**Informed Consent Statement:** Informed consent was obtained from all subjects involved in the study.

**Data Availability Statement:** No data available.

**Conflicts of Interest:** There is no conflict of interest among the authors.

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
