# Peer review of "River Green Land and Its Influence on Urban Economy, Leisure Development, Ecological Protection, and the Well-Being of the Elderly"

_water, doi:10.3390/w15071350_

Round 1
Reviewer 1 Report
The paper is interesting, as you mentioned it is s pity, that your extended ideas for research couldn't be done by your capacity.
Author Response
Reviewer 1
The paper is interesting, as you mentioned it is s pity, that your extended ideas for research couldn't be done by your capacity.
We appreciate your suggestion and approval. We try our best to elaborate and verify this idea. I hope that this manuscript will provide more people to complete this answer after publication.
We are very grateful to the reviewers, and we thank you again for your kind suggestions, which have been revised to the best of our ability. Hope to get your approval, and sincerely look forward to receiving good news.
Reviewer 2 Report
The comments are indicated in the file.
Please indicate the improvements in red.
It needs to be explained. The river valley's ecosystem has been changed until becoming a canal in some places.
In this part, it is necessary to explain what has been done wrong that the river valleys need water conservancy.
Natural river valleys do not need greenery. Please explain why the natural river valley vegetation needs to be changed.
Explain why water engineering technology is necessary. The not modified river valleys are self-sustaining ecosystems and do not need any improvement.
line 76? How is this possible? The ecological processes adjusting to the conditions.
Define green areas.
What can be better than natural ecosystems?
Why is the emergence of green areas necessary? What is wrong with natural river vegetation?
What is the difference between environmental and ecological aspects? Why do you think they need development? Ecosystems develop according to the successional rules.
This shows the influence of river water conservancy projects on villages' economic, social, and environmental development. The passage above is more than justifying this statement.
170 We believe that the impact of river water projects on village development can be understood by exploring villages' economic, social, and environmental aspects. This approach suggests making new mistakes to cover the old ones.
What is the natural capital of the river valley responding? The paper presents an extreme Anthropocentric approach.

Author Response
Reviewer 2
The comments are indicated in the file.
Please indicate the improvements in red.
- Introduction第2段
It needs to be explained. The river valley's ecosystem has been changed until becoming a canal in some places.
Thanks for your suggestion, we have added a description of why the river valley ecosystem was changed and what it is used for.
In this part, it is necessary to explain what has been done wrong that the river valleys need water conservancy.
Thanks to your suggestion, we have revised this narrative and accounted for the original flaws of the hydraulic works. The above description is as in lines 49-72.
Natural river valleys do not need greenery. Please explain why the natural river valley vegetation needs to be changed.
Thanks to your suggestion, we have revised this narrative. Such as lines 68-69.
- Introduction
Explain why water engineering technology is necessary. The not modified river valleys are self-sustaining ecosystems and do not need any improvement.
Thanks to your suggestion, we have revised this narrative. Such as lines 76-78.
line 76? How is this possible? The ecological processes adjusting to the conditions.
Thanks to your suggestion, we have revised this narrative. Such as lines 76-78.
Define green areas.
Thanks for the suggestion, we have defined it on lines 67-68.
What can be better than natural ecosystems?
Thanks to your suggestion, we have revised this narrative. Such as lines 76-78.
Why is the emergence of green areas necessary? What is wrong with natural river vegetation?
Thanks for your suggestion, we bring up the importance of river green space and explain the difference between natural vegetation and artificial green space. Such as lines 86-93.
- Introduction
What is the difference between environmental and ecological aspects? Why do you think they need development? Ecosystems develop according to the successional rules.
Thanks for the suggestion, I'll make a note of it.
The environment referred to in this article refers to the rural community environment, and in addition, what we want to talk about is the effect of ecological conservation. This paragraph we have revised the description and presented in lines 107-110.
2.2
This shows the influence of river water conservancy projects on villages' economic, social, and environmental development. The passage above is more than justifying this statement.
This shows the impact of river water conservancy projects on the economic, social and environmental development of villages. The paragraph above is more than justification for this statement.
2.2
170 We believe that the impact of river water projects on village development can be understood by exploring villages' economic, social, and environmental aspects.
This approach suggests making new mistakes to cover the old ones.
Thanks to your suggestion, we have revised this narrative. Such as 154-158.
What is the natural capital of the river valley responding?
Thank you for your suggestion. The topic of our discussion on the natural environment and ecology of the river is to discuss:
N1: River water quality is turbid (0.863)
N2: Changes in channel soil structure (0.841)
N3: Decrease in the number of ecological species (0.868)
N4: River ecology and natural environment are damaged (0.875)
N5: Addition of tourist garbage (0.855)
We think this can confirm the impact on the original natural environment and ecology after the construction of green space and the use of this space to promote construction and development by the city.
The paper presents an extreme Anthropocentric approach.
Thank you for your suggestions and opinions. In this manuscript, we discuss the effects of river greening on urban development, the effectiveness of river natural environment and ecological conservation, and the well-being of the elderly from the perspective of people.
We are very grateful to the reviewers, and we thank you again for your kind suggestions, which have been revised to the best of our ability. Hope to get your approval, and sincerely look forward to receiving good news.
Reviewer 3 Report
Here are my comments regarding this manuscript:
(1) Title of the paper is too long, and should definitely not be provided in a form of a question. Authors are strongly advised to redefine title of the paper
(2) There are far too many references in the reference list. Authors are advised to reduced this number up to a half of the present number.
(3) I cannot see any scientific novelty in the paper. It seems authors just provide analysis of the answers to the predefined question. Authors are strongly encouraged to provide clear explanation of a scientific contribution of the paper.
Author Response
Reviewer 3
Here are my comments regarding this manuscript:
- Title of the paper is too long, and should definitely not be provided in a form of a question. Authors are strongly advised to redefine title of the paper
Thanks for your suggestion, we have shortened the topic word count and length.
- There are far too many references in the reference list. Authors are advised to reduced this number up to a half of the present number.
Thanks for your suggestion, we have adjusted the number of references.
(3) I cannot see any scientific novelty in the paper. It seems authors just provide analysis of the answers to the predefined question. Authors are strongly encouraged to provide clear explanation of a scientific contribution of the paper.
Thanks for your suggestion, we have reworked the Discussion and Conclusion sections. Such as lines 343-367, and lines 410-433.
We are very grateful to the reviewers, and we thank you again for your kind suggestions, which have been revised to the best of our ability. Hope to get your approval, and sincerely look forward to receiving good news.
Reviewer 4 Report
Authors
I have a concern regarding the structure of the questionnaire and the list of questions. The authors should explain the areas of evaluations, related questions, sampling procedures, and descriptive findings. Why some questions are posed, and how they are related to the topic(s) of the study. Some references to similar studies should be compared in the discussion section. Following are the few corrections needed in the manuscript.
Title
Line no-4: The title is too long. Can you make it short by removing the question? The authors describe the situation of the rural area instead of the Urban. So, please replace the word urban with “rural” in the title.
Abstract
Line no-26: The word “findings” is not necessary for the middle of the abstract. Please remove it.
Literature discussion
Line no-117: Please remove the double dot after section 2.
Methodology
Line no-223: In figure 1. The world ‘eder’ suggested replacing it with “Elder”.
Line no-242: Please try to make all hypotheses statistically proven, i.e. make it all null hypothesis (Ho) or all alternative hypothesis (H1).
Line no-297: Authors are suggested to replace the word ‘manpower’ with "Human Resource".
Results
Line no-342: Authors are requested to make ascending or descending order based on the ‘M’ or “ranking” column.
Line no-366: Please add the title of the column and make the left alignment in table 3.
Discussion:
Line no- 368: Authors are suggested to make a comparison with other evidence.
Recommendation:
Line no- 468: Authors are requested to remove sections 5.1, 5.2, and 5.3 or make it a short paragraph in the conclusion section.
Author Response
Reviewer 4
I have a concern regarding the structure of the questionnaire and the list of questions. The authors should explain the areas of evaluations, related questions, sampling procedures, and descriptive findings. Why some questions are posed, and how they are related to the topic(s) of the study. Some references to similar studies should be compared in the discussion section.
We are very grateful to the reviewers, and we thank you again for your kind suggestions, which have been revised to the best of our ability. Hope to get your approval, and sincerely look forward to receiving good news.
Following are the few corrections needed in the manuscript.
Thank you for your suggestions. We will reply to your suggestions in turn below.
Title
Line no-4: The title is too long. Can you make it short by removing the question? The authors describe the situation of the rural area instead of the Urban. So, please replace the word urban with “rural” in the title.
Thanks for your suggestion, we have adjusted the theme names and replaced urban with rural.
Abstract
Line no-26: The word “findings” is not necessary for the middle of the abstract. Please remove it.
Thanks for your suggestion, we have removed it.
Literature discussion
Line no-117: Please remove the double dot after section 2.
Thanks for your suggestion, we have removed it.
Methodology
Line no-223: In figure 1. The world ‘eder’ suggested replacing it with “Elder”.
Thanks for your suggestion, we have modified it.
Line no-242: Please try to make all hypotheses statistically proven, i.e. make it all null hypothesis (Ho) or all alternative hypothesis (H1).
Thanks for your suggestion, we have modified it. Such as lines 230-237.
Line no-297: Authors are suggested to replace the word ‘manpower’ with "Human Resource".
Thanks for your suggestion, we have modified it.
Results
Line no-342: Authors are requested to make ascending or descending order based on the ‘M’ or “ranking” column.
Thanks for your suggestion, we have modified it. Such as lines 334-338.
Line no-366: Please add the title of the column and make the left alignment in table 3.
Thanks for the suggestion, we've added it. Align the text to the left.
Discussion:
Line no- 368: Authors are suggested to make a comparison with other evidence.
Thanks for your suggestion, we have supplemented it and readjusted the content. Such as lines 343-367.
Recommendation:
Line no- 468: Authors are requested to remove sections 5.1, 5.2, and 5.3 or make it a short paragraph in the conclusion section.
Thanks for your suggestion, we have revised this narrative. Such as lines 410-433.
We are very grateful to the reviewers, and we thank you again for your kind suggestions, which have been revised to the best of our ability. Hope to get your approval, and sincerely look forward to receiving good news.
Round 2
Reviewer 3 Report
Authors did not provide satisfying responses. Novelty of the paper is still not clear. Therefore, I recommend the rejection of the manuscript for publication in its current form.
Author Response
Reviewer 3
First of all, we would like to thank you for your assistance and offer the opportunity to modify. We reply as follows according to your suggestion:
- We have revised the title to shorten its length.
- We also edited the references, kept the necessary references, and reduced the number.
- We have rewritten the abstract, conclusion and other descriptions to strengthen the importance of this article.
- We also asked native English speakers to assist in grammar editing, hoping to enhance the visualization of the manuscript.
We have done our best to make changes and hope to get your approval.

Reviewer 4 Report
I would like to thank the authors for addressing all comments. The authors massively revised the manuscript and make it more understandable. I suggest editing the English language once.
Author Response
Reviewer 4
First of all, we would like to thank you for your assistance and offer the opportunity
to modify. We reply as follows according to your suggestion:
We asked native English speakers to assist in grammar editing, hoping to enhance the
visualization of the manuscript.
We have done our best to make changes and hope to get your approval.
